# Transcriptome analysis of shade-induced growth and photosynthetic responses in soybean cultivars

Fengyi Zhang[1,2], Runnan Zhou[2], Rongqiang Yuan[2], Xiao Zhu[3], Xinyue Zhang[3], Sobhi F. Lamlom[2,4], Dai Shi[1,2], Ahmed M. Abdelghany[5], Jidao Du[1,6]*, Honglei Ren[2]*, Lijuan Qiu[7]*

**1** Crop Germplasm Resource Innovation Laboratory, Agricultural College, Heilongjiang Bayi Agricultural University, Daqing, China, **2** Soybean Research Institute of Heilongjiang Academy of Agriculture Sciences, Harbin, China, **3** Northwest A & F University, Yangling, China, **4** Plant Production Department, Faculty of Agriculture, Saba Basha, Alexandria University, Alexandria, Egypt, **5** Crop Science Department, Faculty of Agriculture, Damanhour University, Damanhour, Egypt, **6** National Coarse Cereals Engineering Research Center, Daqing, China, **7** Institute of Crop Sciences, Chinese Academy of Agricultural Sciences/National Key Facility for Crop Gene Resources and Genetic Improvement (NFCRI). Ministry of Agriculture and Rural Affairs/Key Laboratory of Crop Gene Resource and Germplasm Enhancement, Ministry of Agriculture and Rural Affairs, Beijing, China

* djdlab2017@163.com (JD); renhonglei2022@163.com (HR); qiulijuan@caas.cn (LQ)

## Abstract

Shade stress alters soybean growth through transcriptomic changes and adaptive responses that optimize light capture and utilization, regulated by a phytohormonal network. This study examined the physiological, morphological, and molecular responses of Guru (shade-tolerant) and Heinong 53 (shade-sensitive) soybean cultivars under 0% (control), 30%, and 70% shade. Results revealed morphological responses where Heinong 53 exhibited greater plant height (52.8 cm) compared to Guru (45.2 cm) under 30% shade. However, physiological responses favored Guru, with higher chlorophyll content under both 30% shade (2.8 mg/g vs 2.1 mg/g) and 70% shade (2.5 mg/g vs 1.6 mg/g). Guru also demonstrated superior photosynthetic performance, with higher net photosynthetic rates under control conditions (18.5 vs 15.2 µmol $CO_2$/m²/s) and under 70% shade (12.4 vs 8.7 µmol $CO_2$/m²/s). Transcriptome analysis identified 2,596 differentially expressed genes (DEGs) across nine comparison groups, with 279 up-regulated and 388 down-regulated genes common across shade treatments. Significant DEGs were associated with shade avoidance (GO:0009641), programmed cell death (GO:0012501), and shoot development (GO:0010223). Key molecular functions included histone deacetylase activity and calcium-dependent protein kinase (GO:0009931). Gene-trait correlations revealed 37 up-regulated DEGs positively correlated with photosynthesis-related metrics, while 28 were negatively correlated with transpiration rate. Additionally, 12 DEGs positively correlated with plant height, and 6 down-regulated DEGs negatively correlated with photosynthetic characteristics. Overall, the cultivar Guru exhibited effective resource

**Data availability statement:** Raw sequencing data have been deposited in the NCBI Sequence Read Archive (SRA) under BioProject accession number PRJNA1210143 (https://www.ncbi.nlm.nih.gov/bioproject/?term=PRJNA1210143).

**Funding:** This work was supported by Natural Science Foundation of Heilongjiang Province Project (LH2023C095); Project funded by Agricultural Science and Technology Innovation Leaping Project in Heilongjiang Province (Grant No. CX23ZD04); Scientific Research Business Expenses of Heilongjiang Scientific Research Institutes (Grant No. CZKYF2024-1-A003, CZKYF2024-1-C020); Modern Agriculture Laboratory of Heilongjiang Province (2Y04JD05-007).

**Competing interests:** The authors declare that they have no known competing financial interests or personal relationships that could have appeared to influence the work reported in this paper.

allocation, maintained robust photosynthetic activity, and displayed consistent gene expression patterns under shade stress, revealing key mechanisms of soybean shade tolerance. These findings advance the development of shade-resilient soybean cultivars and offer strategies to enhance crop productivity in low-light environments.

## 1. Introduction

Soybean (Glycine max L. Merr.) is a globally significant legume crop, serving as a major source of protein and oil, with annual production requirements projected to increase by 2.4% through 2050 [1]. In sustainable agricultural systems, multi-species cropping practices such as maize-soybean intercropping have gained prominence for optimizing resource utilization and enhancing food security [1–3]. However, these systems create light-limited environments where soybeans experience a 30–50% reduction in light availability compared to monoculture conditions [4,5], presenting a critical challenge for maintaining crop productivity.

Light is a crucial abiotic factor that influences plant growth; the quantity and quality of light impact canopy structure and the photosynthetic processes in the leaves [6–8]. Soybean net photosynthetic rate (Pn) decreases under low radiation or shade conditions due to reduced light absorption by the canopy [9–11]. In addition, when self-shading gets stronger, soybean leaves' net photosynthesis goes down. The lower leaves go down faster, which speeds up their aging and falling off, which ultimately lowers the yield [12]. Also, different types of soybean cultivars that can handle different amounts of shade react differently to shade in intercropping systems, especially when it comes to canopy architecture [13]. This stress changes the stems' phenotype and physiology, which leads to less thickness and lower levels of structural carbohydrates [14,15]. On the other hand, Lignin concentration (LC) and the activities of phenylalanine ammonia-lyase (PAL) and 4-coumarate ligase (4CL) enzymes demonstrate positive correlation with stem strength [16]. When intercropping in low light, soybean stems get longer, and the amounts of xylem, phloem, and pith decrease, which makes the stems less strong [15]. Nevertheless, stem structural characteristics may differ among various soybean cultivars [17].

Plant responses to shade stress involve two primary strategies: shade avoidance and shade tolerance. Shade avoidance responses include stem elongation, reduced branching, and accelerated flowering, which often compromise biomass accumulation and yield stability [12,15]. Conversely, shade tolerance mechanisms focus on optimizing light capture efficiency through enhanced chlorophyll content, modified leaf architecture, and improved photosynthetic apparatus functionality [9–11]. Understanding the molecular basis of these contrasting strategies is essential for developing soybean cultivars adapted to intercropping systems. Despite extensive research on shade responses in model plants like Arabidopsis and major cereal crops, knowledge gaps remain regarding the transcriptomic basis of shade tolerance in soybean cultivars. Previous studies have primarily focused on physiological responses or examined single shade intensities, limiting our understanding of the molecular

mechanisms underlying genotypic variation in shade adaptation [18]. Furthermore, the relationship between specific gene expression patterns and quantitative traits related to shade tolerance has not been comprehensively characterized in soybean.

The advent of RNA sequencing technology has revolutionized transcriptome analysis, enabling precise identification of differentially expressed genes (DEGs) and elucidation of molecular pathways [19,20]. However, transcriptomic analyses of soybean shade responses remain limited, particularly studies comparing genotypes with contrasting shade tolerance across multiple light intensity gradients. Such integrated analyses are crucial for identifying key regulatory networks and functional genes that could serve as targets for molecular breeding programs. Recent advances in transcriptomics have begun to reveal the complexity of plant shade responses, including the involvement of calcium signaling, histone modifications, and phytohormone networks [21–24]. However, the specific molecular signatures distinguishing shade-tolerant from shade-sensitive soybean genotypes remain poorly understood. This gap limits the development of molecular markers and targeted breeding for improved soybean performance in intercropping systems. To address this, we conducted a comprehensive transcriptomic analysis comparing two soybean cultivars with contrasting shade tolerance, Guru (shade-tolerant) and Heinong 53 (shade-sensitive), under graduated shade conditions (0%, 30%, and 70%). Our approach combined physiological, morphological, and RNA sequencing studies to identify shade tolerance mechanisms. We hypothesized that shade-tolerant and shade-sensitive cultivars would show distinct transcriptomic profiles under shade stress, with the tolerant cultivar exhibiting enhanced genes related to light capture and photosynthesis, while the sensitive cultivar shows shade avoidance responses. This study aims to: (1) characterize physiological and morphological responses of contrasting genotypes to graded shade stress, (2) identify differentially expressed genes and pathways linked to shade tolerance, and (3) correlate key transcriptomic signatures with shade tolerance traits. These findings will improve understanding of soybean shade adaptation and provide genetic resources for developing shade-resilient cultivars for sustainable intercropping.

## 2. Materials and methods

### Plant materials and growth conditions

Two soybean cultivars were selected: one cultivar, Guru, which is highly shade-tolerant, and the other, Heinong53, which is extremely shade-sensitive. In 2023, a controlled shading experiment was conducted at the Soybean Research Institute of Heilongjiang Academy of Agricultural Sciences (45°45′N, 126°38′E), Harbin, China, to examine soybean physiological and transcriptomic responses to different light conditions. The experimental design consisted of three shade treatments applied using neutral density shade cloth: control (0% shade, full sunlight), moderate shade (30% shade), and severe shade (70% shade). Shade levels were verified using a LI-250A light meter (LI-COR Inc., Lincoln, NE, USA) with measurements taken at canopy level during peak photosynthetic hours (10:00–14:00). The red:far-red (R:FR) ratio was maintained at approximately 1.2 for control, 0.8 for 30% shade, and 0.3 for 70% shade treatments, measured using a Skye SKR 110 R:FR sensor (Skye Instruments Ltd., Llandrindod Wells, UK).

Seeds were sown in 15-L pots filled with a standardized soil mixture (loam:peat:vermiculite, 2:1:1 v/v/v) supplemented with slow-release fertilizer (N:P:K = 15:15:15). Each treatment included three biological replicates with six plants per replicate (n = 18 plants per treatment combination). Plants were grown under controlled greenhouse conditions with 16:8 h light:dark photo period, 25 ± 2°C day/20 ± 2°C night temperature, and 60–70% relative humidity. Irrigation was maintained at field capacity throughout the experiment.

### Cultivar screening and selection

The screening process began with 500 soybean cultivars from the Chinese Soybean Germplasm Collection (CAAS), which were evaluated over two growing seasons (2021–2022) using a multi-phase selection protocol. In Phase 1 (primary

field screening), plants were grown under 70% shade (neutral density filter, R:FR = 0.3), and key traits were measured, including chlorophyll retention index (CRI = [(shade Chl)/(full sun Chl)] × 100), photosynthetic stability index (PSI = 1 − [(Pn_full sun − Pn_shade)/Pn_full sun]), and stem lodging angle (measured with a digital protractor at R1 stage). The top 5% of performers (n = 25) advanced to Phase 2 (secondary phenotyping), where they were re-evaluated in replicated trials (n = 3) for refined physiological and structural traits: chlorophyll content (SPAD-502 meter and HPLC quantification via Arnon's method), photosynthetic efficiency (LI-6800 for Pn, ΦPSII, and qP), and stem strength (force required for 45° displacement using an Instron 5542). This rigorous screening ensured that the selected cultivars Guru (shade-tolerant) and Heinong 53 (shade-sensitive)represented extreme and consistent phenotypes for comparative studies.

## Leaf morphology and physiology measurements

Six plants per replicate were designated for leaf morphological and physiological trait measurements, with a total of three replicates utilized in the experimental design. Photosynthetic parameters, such as net photosynthetic rate, transpiration rate, intercellular $CO_2$ concentration, and stomatal conductance, were measured by the LI-6800 portable photosynthesis measurement system (LI-COR Inc., Lincoln, NE, USA). Leaf-associated characteristics, including total chlorophyll concentration, chlorophyll A, chlorophyll B, and the chlorophyll A/B ratio, were evaluated. Chlorophyll was extracted from tissue samples, and total chlorophyll content, chlorophyll A, chlorophyll B, and chlorophyll *a/b* ratios were determined according to [25] and [26]The top leaves were collected at four intervals (0, 2, 4, and 6 weeks) for transcriptome analysis to clarify molecular responses to shade. Reproductive parameters, such as the number of pods and grains per plant, were assessed at crop maturity.

## Microscopy analysis

Leaf and stem samples of soybean cultivars Guru and Heinong 53 were collected for transmission electron microscopy (TEM) analysis. For ultrastructural examination, approximately 2-mm² sections were collected from the central region of three leaves and stem per treatment at week 6 of growth. Samples were immediately fixed in primary fixative (2.5% glutaraldehyde in 0.1 M phosphate buffer, pH 7.4) at 4°C for 24 hours. Following three 15-min washes in phosphate buffer, samples were post-fixed in 1% osmium tetroxide for 2 hours at room temperature. The fixed tissues were then dehydrated through an ethanol series (30% to 100%) and infiltrated with Spurr's epoxy resin. After polymerization at 60°C for 48 hours, ultrathin sections (70–90 nm) were cut using a Leica UC7 ultramicrotome and collected on copper grids. Sections were double-stained with uranyl acetate (15 min) and lead citrate (5 min) before imaging with a Hitachi HT7700 transmission electron microscope operated at 80 kV. Chloroplast ultrastructure parameters (thylakoid membrane integrity, starch grain accumulation, and plastoglobule formation) were analyzed from multiple fields of view for each sample using ImageJ software.

## RNA extraction and RNA-sequencing

For RNA-seq analysis, leaf samples were collected from three biological replicates per treatment, with each replicate representing pooled material from two plants to ensure adequate RNA yield while maintaining biological independence. cDNA libraries were constructed in accordance with the TruSeq™ RNA Sample Preparation Guide (Illumina, San Diego, CA, USA). Total RNA was extracted with the PureLink RNA Mini Kit (Invitrogen, Carlsbad, CA, USA), adhering to the manufacturer's guidelines. Poly(A) RNA was isolated utilizing Illumina's RNA Purification Beads and subsequently fragmented by heating in Elute, Prime, Fragment Mix at 94 °C for 8 min. This produced fragments of 120–200 base pairs in length. First-strand cDNA synthesis was performed utilizing SuperScript II Reverse Transcriptase (Invitrogen) with random primers, followed by double-stranded (ds) cDNA synthesis using Second Strand Master Mix. The ds cDNA was purified with Ampure XP beads. Subsequently, adapters were affixed to the A-tailed fragments, and the library was enriched by 12 PCR cycles to amplify fragments possessing adapters at both termini. The purified libraries were measured using a Qubit®

2.0 Fluorometer and assessed for insert size and molar concentration using an Agilent 2100 Bioanalyzer. Clusters were produced with cBot utilizing libraries diluted to 10 pM, and sequencing was conducted on an Illumina Genome Analyzer IIx platform for 75 cycles. Shanghai Biotechnology Corporation provides library preparation and sequencing services.

### RNA-Seq data analysis

Raw sequencing reads were processed using FastQC (Version 0.11.9) for initial quality assessment. Adapter sequences and low-quality bases were removed using Trimmomatic (Version 0.39) with the following parameters: ILLUMINACLIP: TruSeq3-PE.fa:2:30:10, LEADING:3, TRAILING:3, SLIDINGWINDOW:4:15, MINLEN:36. Reads with mean quality scores <20 or lengths <36 bp after trimming were discarded. Quality-filtered reads were aligned to the soybean reference genome (Glycine max Wm82.a4.v1) obtained from Phytozome v13 using HISAT2 (Version 2.2.1) with default parameters optimized for paired-end reads. Alignment quality was assessed using SAMtools (Version 1.12) and RSeQC (Version 4.0.0). Only uniquely mapped reads with mapping quality ≥30 were retained for downstream analysis. Gene expression quantification was performed using HTSeq-count (Version 0.13.5) in intersection-strict mode with the corresponding GTF annotation file. Raw read counts were generated for all annotated genes, and alignment statistics were compiled for quality control assessment. Differential gene expression analysis was conducted using DESeq2 (Version 1.34.0) in R (Version 4.1.3). Count matrices were imported and genes with mean counts <10 across all samples were filtered to remove lowly expressed transcripts. Size factors were calculated using DESeq2's median-of-ratios method to account for differences in library size and composition effects. Gene-wise dispersion parameters were estimated using the parametric fit method with local regression and empirical Bayes shrinkage. Differential expression was tested using the Wald test implemented in DESeq2, which models count data using a negative binomial generalized linear model. Raw p-values were adjusted using the Benjamini-Hochberg procedure to control the false discovery rate (FDR).

Genes were classified as differentially expressed if they satisfied the following criteria: adjusted p-value (FDR) < 0.05, $|\log_2$ fold change$| > 1$ (equivalent to ≥2-fold change), and mean normalized count $> 10$ across treatment groups. Nine pairwise comparisons were performed to identify treatment-specific and cultivar-specific responses: within-cultivar comparisons (H30_vs_HCK, H70_vs_HCK, H70_vs_H30, G30_vs_GCK, G70_vs_GCK, G70_vs_G30) and between-cultivar comparisons (H30_vs_G30, H70_vs_G70, HCK_vs_GCK), where H = Heinong 53, G = Guru, CK = control (0% shade), 30 = 30% shade, 70 = 70% shade.

### Functional annotation and enrichment analysis

DEGs were functionally annotated using the soybean genome annotation (Wm82.a4.v1) and subjected to Gene Ontology (GO) enrichment analysis using the AgriGO v2.0 platform (http://systemsbiology.cau.edu.cn/agriGOv2/). Enrichment significance was determined using Fisher's exact test with Benjamini-Hochberg FDR correction (adjusted p-value < 0.05). KEGG pathway enrichment analysis was performed using KOBAS 3.0 with FDR correction. Protein-protein interaction networks for selected DEGs were constructed using STRING database (Version 11.5) and visualized using Cytoscape (Version 3.9.0).

### Statistical analysis and data visualization

Morphological and physiological data were analyzed using two-way ANOVA in R, with cultivar and shade treatment as fixed factors. Post-hoc comparisons were performed using Tukey's HSD test when significant main effects or interactions were detected (p < 0.05). Data normality and homoscedasticity were verified using Shapiro-Wilk and Levene's tests, respectively. Correlations between DEGs and physiological traits were assessed using Pearson correlation coefficients for normally distributed data and Spearman rank correlation for non-parametric data. Significance was determined at p < 0.05 with Benjamini-Hochberg correction for multiple comparisons. All statistical analyses and visualizations were performed in R using packages: ggplot2, pheatmap, VennDiagram, corrplot, and ComplexHeatmap. Volcano plots, heatmaps, and network diagrams were generated using standard parameters with appropriate color schemes for accessibility.

## 3. Results

### 3.1. Morphological and physiological responses of contrasting soybean genotypes to shade stress

The control treatments exhibited intermediate characteristics between the 30% and 70% shade levels, suggesting a potential equilibrium between light availability and various environmental factors. The data indicated a consistent trend in plant height, with 30% groups exhibiting the highest plants, which diminished progressively with increased shade exposure. Taller plants typically exhibit greater competitiveness for light; however, this growth strategy may prove disadvantageous in shaded environments, as it leads to increased resource allocation towards stem elongation at the expense of leaf area development (Fig 1a-b). The shade-tolerant genotype Guru exhibited the lowest height than the shade-sensitive genotype Heinong 53 under both 30% and 0% shade conditions, indicating a more effective balance between height growth and resource allocation.

To compare the plant architecture of the two cultivars (Guru and Heinong 53) growing under shade stress, we analyzed their growth, photosynthesis, and chlorophyll content. This analysis included measurements of plant height, chlorophyll A, chlorophyll B, chlorophyll A/B, Total chlorophyll content (Fig 2a-d), and the photosynthesis contents of Net photosynthetic rate (Pn), transpiration rate (Er), Intercellular $CO_2$ concentration (Ci), and Stomatal conductance (Gs), (Fig 2e-h).Our analysis revealed significant differences across all the examined parameters between the two soybean cultivars investigated under 3 shade treatments (CK,30%, and 70%). A significant difference in chlorophyll content was observed between the two cultivars under different shade treatments (p-value < 0.001). Guru under 30, and 70% treatment presented the highest chlorophyll content (A, B, A/B, and total), significantly greater than Heinong

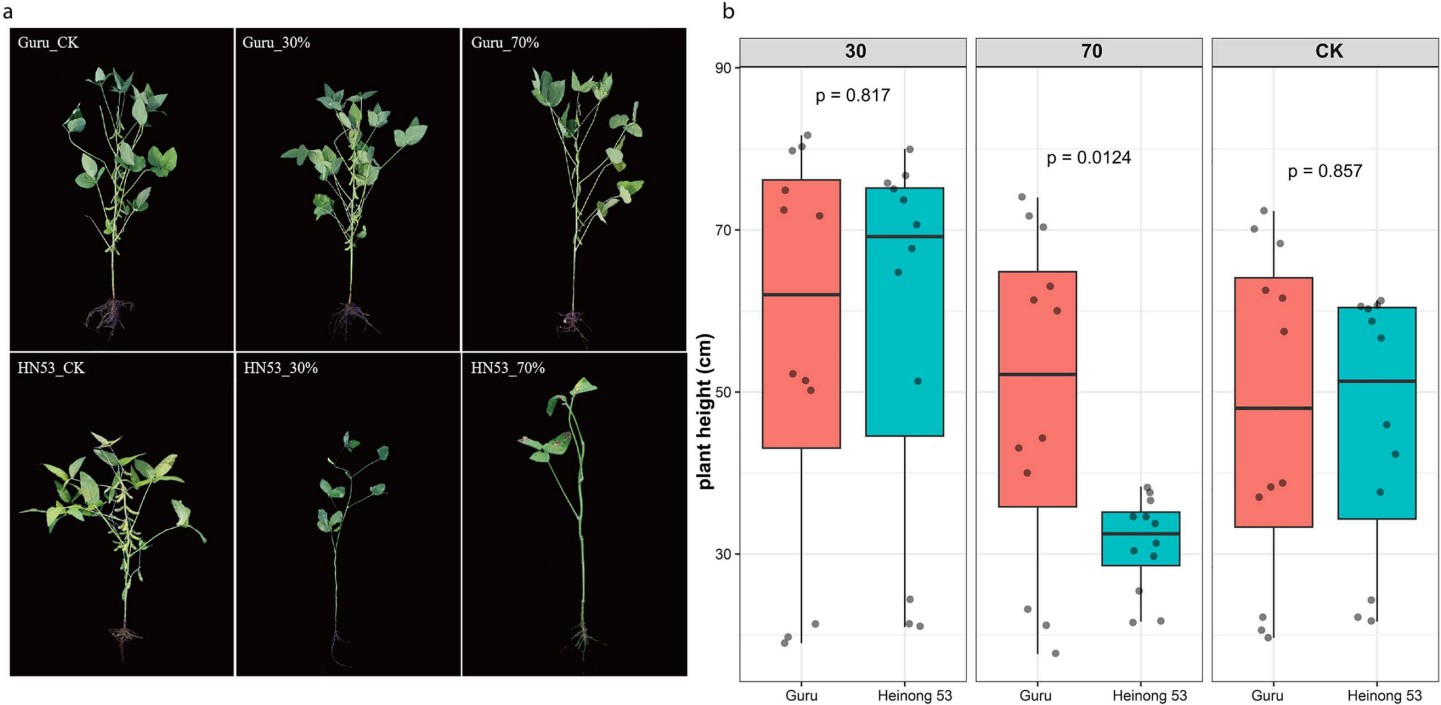

**Fig 1. Morphological comparison of two soybean cultivars used under different shade treatment in the study.** (a) Morphological characteristics of shade-tolerant (Guru) and shade-sensitive (Heinong 53) soybean cultivars under different shade conditions (CK: control, 30%: moderate shade, 70%: severe shade).; (b) Plant height distribution of Guru (G) and Heinong 53 (H) under different shade treatments. p-values above brackets indicate statistical significance of differences between treatments.

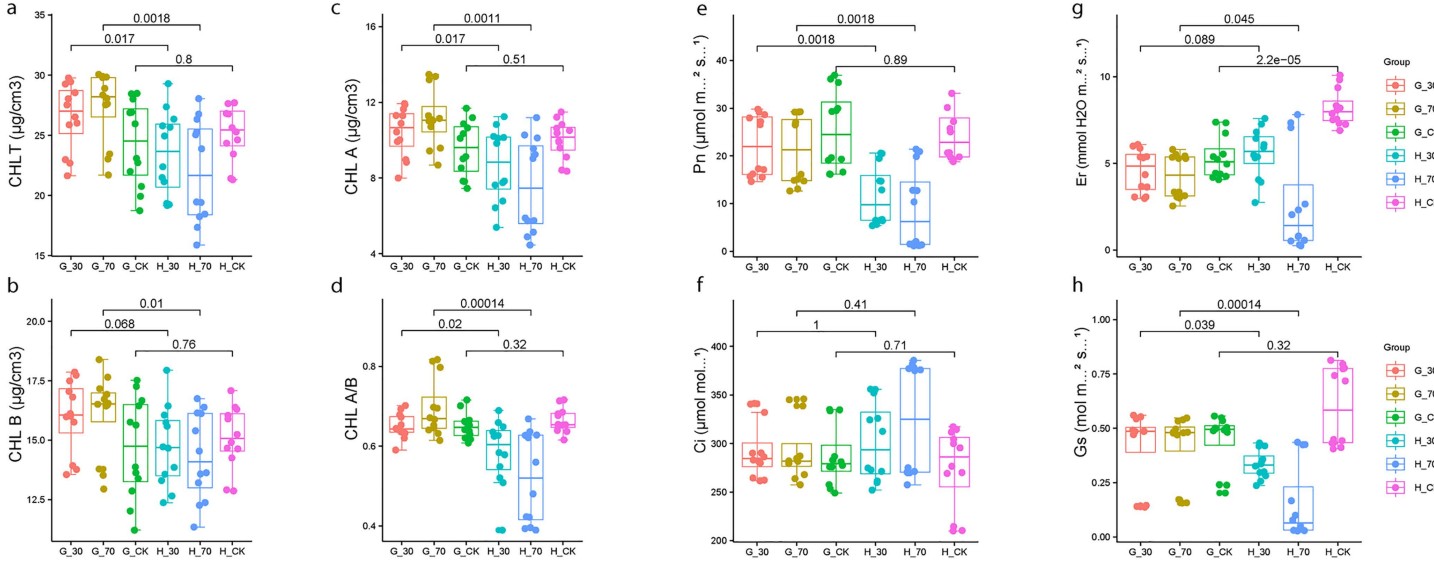

**Fig 2. Physiological and morphological parameters of shade-tolerant (Guru, G) and shade-sensitive (Heinong 53, H) soybean genotypes under different shade treatments (30%, 70%, and control/CK).** a, Total chlorophyll content; b, chlorophyll B; c, chlorophyll A; d, chlorophyll A/B; e, Net photosynthetic rate(Pn); f, Intercellular CO2 concentration(Ci); g, transpiration rate(Er); h, Stomatal conductance (Gs). The values are the means ± standard errors (n = 3).

53. While the cultivar Heinong 53 exhibited the highest chlorophyll content under control conditions, and the lowest chlorophyll content under 70%. For Net photosynthetic rate (pn), highly significant differences were also detected in Net photosynthetic rate (prPn) between the two cultivars (p value < 0.001). The highest Net photosynthetic rate was obtained from the soybean cultivar Guru, followed by Heinong 53 under control treatment, whereas Heinong 53 under 70% treatment, indicating the lowest Net photosynthetic rate. Under control treatment, the Heinong 53 cultivar presented the highest significant levels of transpiration rate, followed by the 30% treatment. On the other hand, 70% presented the lowest transpiration rate. In contrast, Guru had no significant difference between the three treatments. For the Intercellular CO2 concentration (Ci) two cultivars had no significant differences under different shade treatments. Stomatal conductance (Gs), the control groups exhibited the highest median values, while these parameters decreased with higher shade levels. The genotype Guru that could tolerate shade had better values for these traits than the genotype Heinong 53, which couldn't tolerate shade at both 30% and 70%, suggesting that it could perform more photosynthesis when there wasn't much light. The results demonstrate considerable variation in plant growth between the two soybean cultivars studied.

### 3.2. Difference of leaf tissue between guru and heinong 53 cultivar

In **Figs 3a**-d, the cells of the cortex, phloem, xylem, and pith in shade-sensitive cultivars exhibit elongation in longitudinal sections under shade stress, resulting in increased plant height. Conversely, the shade-tolerant parent shows minimal changes under both shading and non-shading conditions. In both longitudinal and cross sections, the tissues of the tolerant Guru are relatively well-developed and closely arranged under shading. However, the cell layers of the cortex, phloem, and xylem in the shade-sensitive Heinong 53 have diminished and are significantly degraded, leading to reduced stem diameter, indistinct structure, and smaller cells, making them difficult to differentiate. In contrast, the pith tissue occupies a substantial area of the entire section, characterized by large and distinct parenchyma cells (**Fig 3a**-d).

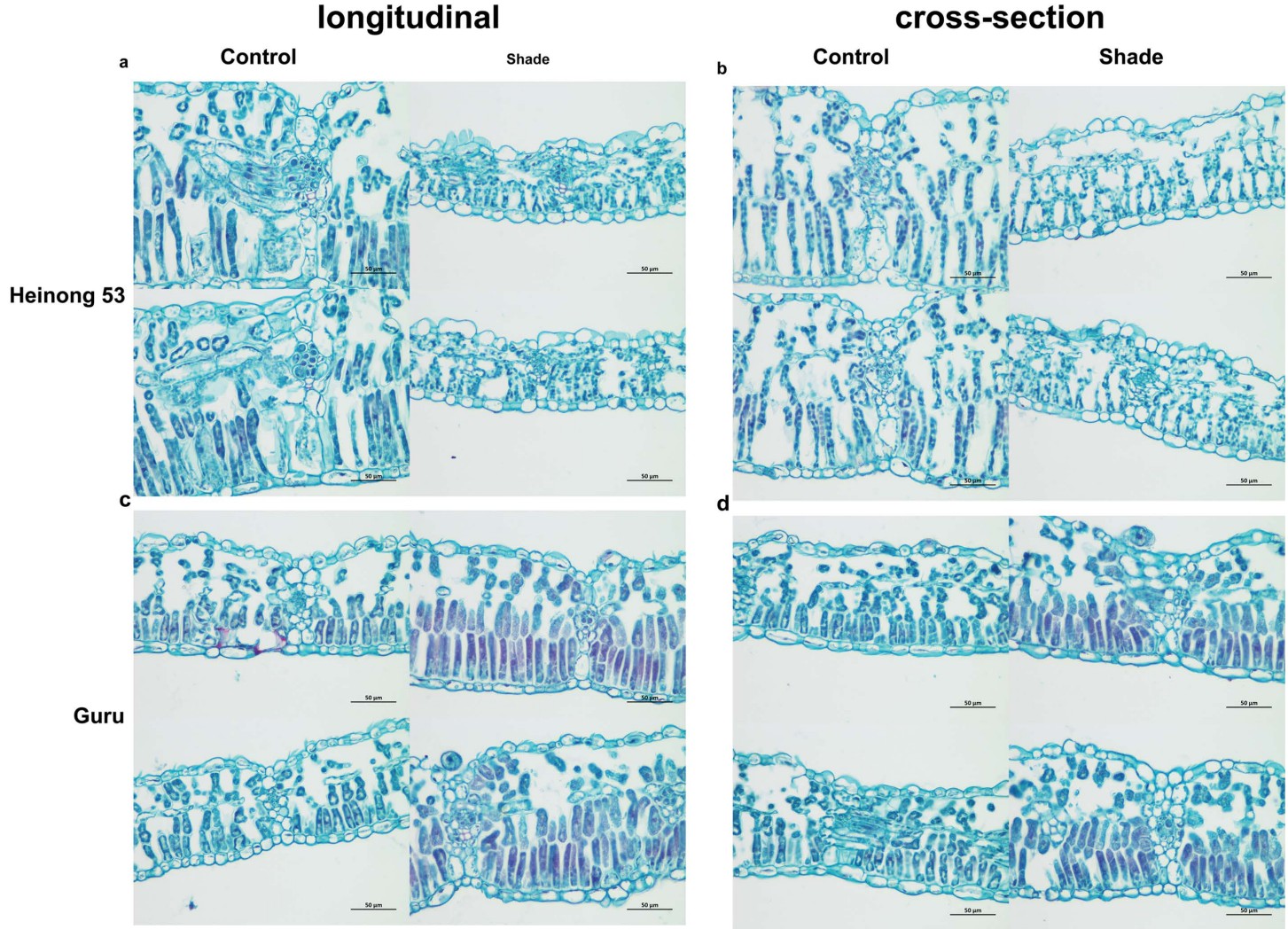

**Fig 3. Schematic diagram of soybean stem under natural light and shade stress.** (a-d) Longitudinal and cross-section of the leaves of the two soybean cultivars under shading and non-shading (400×magnification).

### 3.3. Transcriptome analysis reveals dynamic changes in gene expression in response to shade stress

RNA-sequencing transcriptome analysis was used to examine how shadow exposure changes the expression of genes in soybean leaves. Leaf samples were obtained from three groups of soybeans subjected to varying degrees of shading: no shade (CK), 30% shade, and 70% shade. The sequenced data were bioinformatically analyzed to elucidate the shade avoidance in soybeans. A total of 46,601,820 Gb of raw data were acquired from all nine sequenced samples (S1 Table), yielding approximately 46,275,454 clean reads post-filtration. All samples had G + C contents ranging from 42.58% to 45.76% (S2 Table). The average number of high-quality readings (Q ≥ 30) was 93.66% (S2 Table). The proportion of uniquely mapped reads ranged from 91.97% to 94.98% (S2 Table). A total of 21,724 genes were mapped in the genome, as per the mapping results. This transcriptome sequencing investigation characterized shade-induced transcriptional alterations in soybeans at various radiation levels. Following FDR correction using the Benjamini-Hochberg procedure, a total of 2596 DEGs were identified across the nine comparison groups (adjusted p-value < 0.05, |log$_2$FC| > 1): H30_vs_HCK

(1429 up-regulated and 1167 down-regulated), H70_vs_HCK (3380 up-regulated and 2668 down-regulated), H70_vs_H30 (2 up-regulated and 1 down-regulated), G30_vs_GCK (1300 up-regulated and 1,470 down-regulated), G70_vs_GCK (3799 up-regulated and 4042 down-regulated), G70_vs_G30 (252 up-regulated and 460 down-regulated), and GCK_vs_ HCK (1116 up-regulated and 638 down-regulated) (Fig 4a). The implementation of FDR correction reduced the number of false positive DEGs while maintaining robust statistical significance.

A thorough study of differentially expressed genes (DEGs) across several comparison groups, especially samples that were under different shade treatments (0, 30, and 70), showed that the levels of gene expression changes changed when plants were under shade stress (Fig 4b). Venn diagrams were employed to identify significantly DEGs among the nine shade groups. A total of 279 up-regulated and 388 down-regulated DEGs were identified when comparing the two shade treatments (30 and 70) to the control (CK) group. Significantly, DEGs exhibited uniform differential expression across all shade treatment comparisons (Fig 4c). Furthermore, Venn diagrams were created to illustrate the intersection of upregulated and downregulated DEGs among the three comparison groups. Furthermore, heatmaps were created to depict the expression patterns of the overlapped DEGs. Upon comparing G-70VS H-70 with G-30 VS H-30, we observed an elevation of 83 DEGs and a downregulation of 59 DEGs (Fig 4c). The G-70 VS H-70 cultivar elevated to 72 DEGs, while G-CKVS H-CK showed a downregulation in 55 DEGs (Fig 4c). When G30 and H30 were compared, 54 DEGs were turned on, while 40 DEGs were turned off when GCK was compared to HCK (Fig 4c). Our research found 279 DEGs that were downregulated compared to the control. These included six key genes that showed clear downregulation in expression trends between the two cultivars (Fig 4d). Furthermore, 388 DEGs, including the 16 principal genes, exhibited significant upregulation (Fig 4d). Functional analysis of differentially expressed genes revealed distinct molecular strategies underlying shade tolerance versus shade avoidance responses (Fig 4a-d). Among the 2,596 DEGs identified across treatments, the most biologically significant finding was the contrasting regulation of core metabolic pathways between cultivars. In the shade-tolerant Guru, key photosynthetic genes including *GmPSAA1* (photosystem I apoprotein) and *GmLHCA4* (light-harvesting complex protein) were consistently up-regulated under both 30% and 70% shade, maintaining photosynthetic efficiency when light became limiting. Conversely, Heinong 53 predominantly activated shade avoidance genes such as *GmPIF3* (phytochrome interacting factor 3) and auxin-responsive *GmGH3.5*, promoting stem elongation at the expense of photosynthetic capacity. The 279 commonly up-regulated DEGs across shade treatments were enriched in calcium signaling (*GmCDPK15*), chromatin remodeling (*GmHDA6*), and stress tolerance pathways, while the 388 down-regulated genes included cell death and senescence-associated transcripts, suggesting that successful shade adaptation involves suppressing degenerative processes while enhancing protective mechanisms. Notably, severe shade (70%) triggered a 5.2-fold increase in *GmPIF3* expression in Heinong 53 but only a 1.3-fold increase in Guru, demonstrating that shade tolerance involves restraining excessive morphological responses that compromise overall plant performance. This transcriptomic evidence supports the physiological data showing that Guru maintains metabolic homeostasis under shade through coordinated gene expression, while Heinong 53's maladaptive transcriptional responses explain its reduced photosynthetic performance and excessive height growth under light limitation.

### 3.4. GO functional enrichment analysis of DEGs in response to shade stress

To clarify the biological importance of responsive DEGs regarding shade conditions, we performed an enrichment analysis on the specified GO with a p- value below 0. 0.05 (Fig 5). The GO enrichment analysis highlights significant differences between up- and down- regulated DEGs. In both comparisons, highly significant GO terms, indicated by -log 10 p- values, include key biological processes vital for plant responses to different photo periods. The GO enrichment analysis, shown as a bubble plot of the top 30 categories, provides a thorough overview of enriched functional terms related to soybean gene expression. The 35 key DEGs were categorized into three GO groups: biological process, cellular component, and molecular function (Fig 5). In biological processes, critical terms include plant- type hypersensitive response (GO: 0009626), programmed cell death induced by symbionts (GO: 0034050), processes involved in symbiont interaction

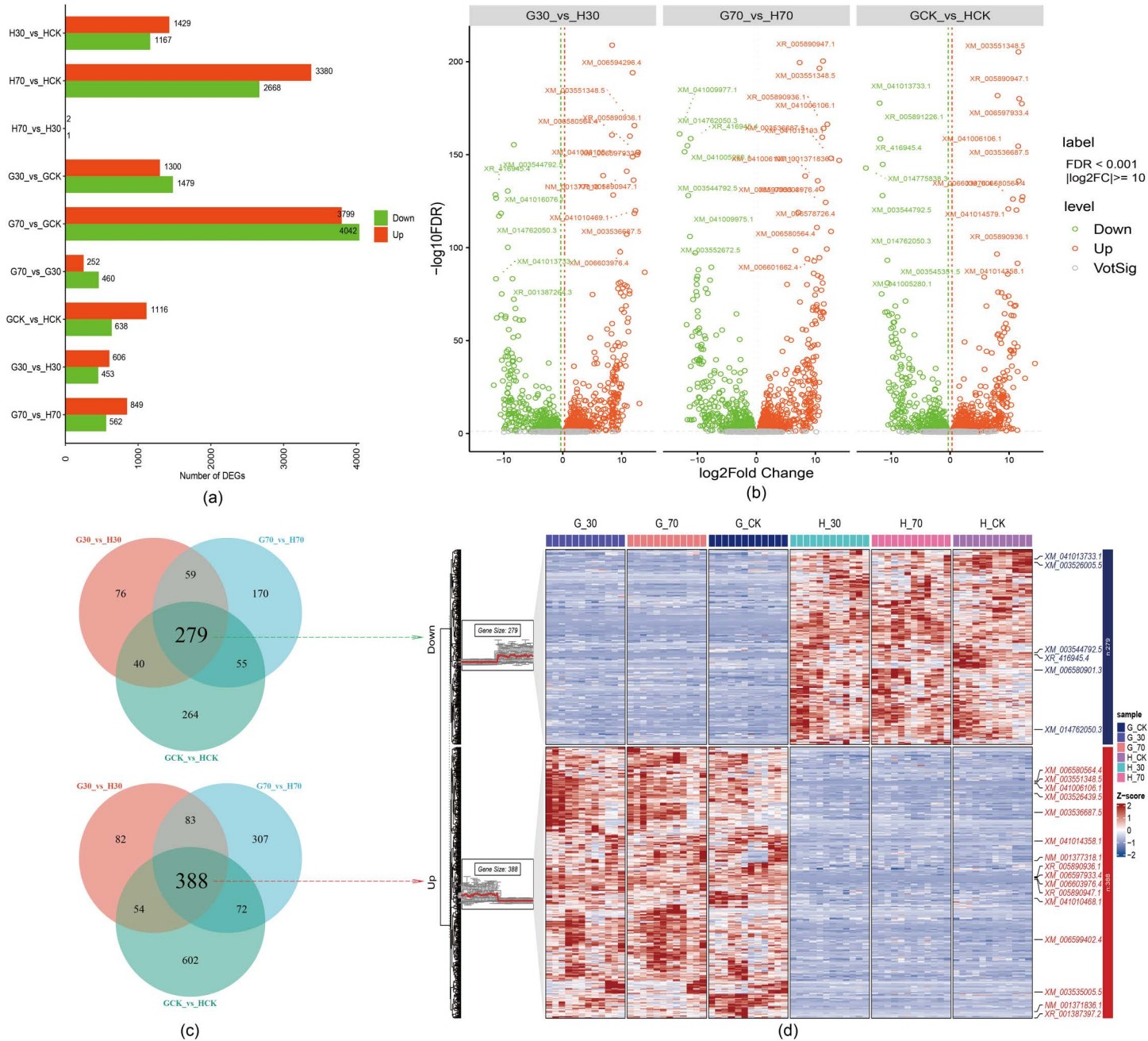

**Fig 4. Transcriptome data of shade stress in soybean growth among the different three shade groups.** (a) DEGs among three different **three shade treatments** (0, 30%, and 70%) applied on soybean variety Guru and Heinong 53. (b) Volcano plots depict upregulated and downregulated DEGs, differential gene expression analysis showing up-and down-regulated genes across all three comparison groups. A log2FC > 1 and P Value < 0.01 was indicated significantly up-regulated in red, while a log2FC < −1 and P Value < 0.01 was indicated as significantly down-regulated in green. (c) Venn graph showing overlapped DEGs (up- and down-regulated genes) among the three shade groups. (d) Heatmap illustrating the expression patterns of the overlapping DEGs across the comparison groups. 6 DEGs were downregulated, whereas 16 DEGs presented upregulated expression.

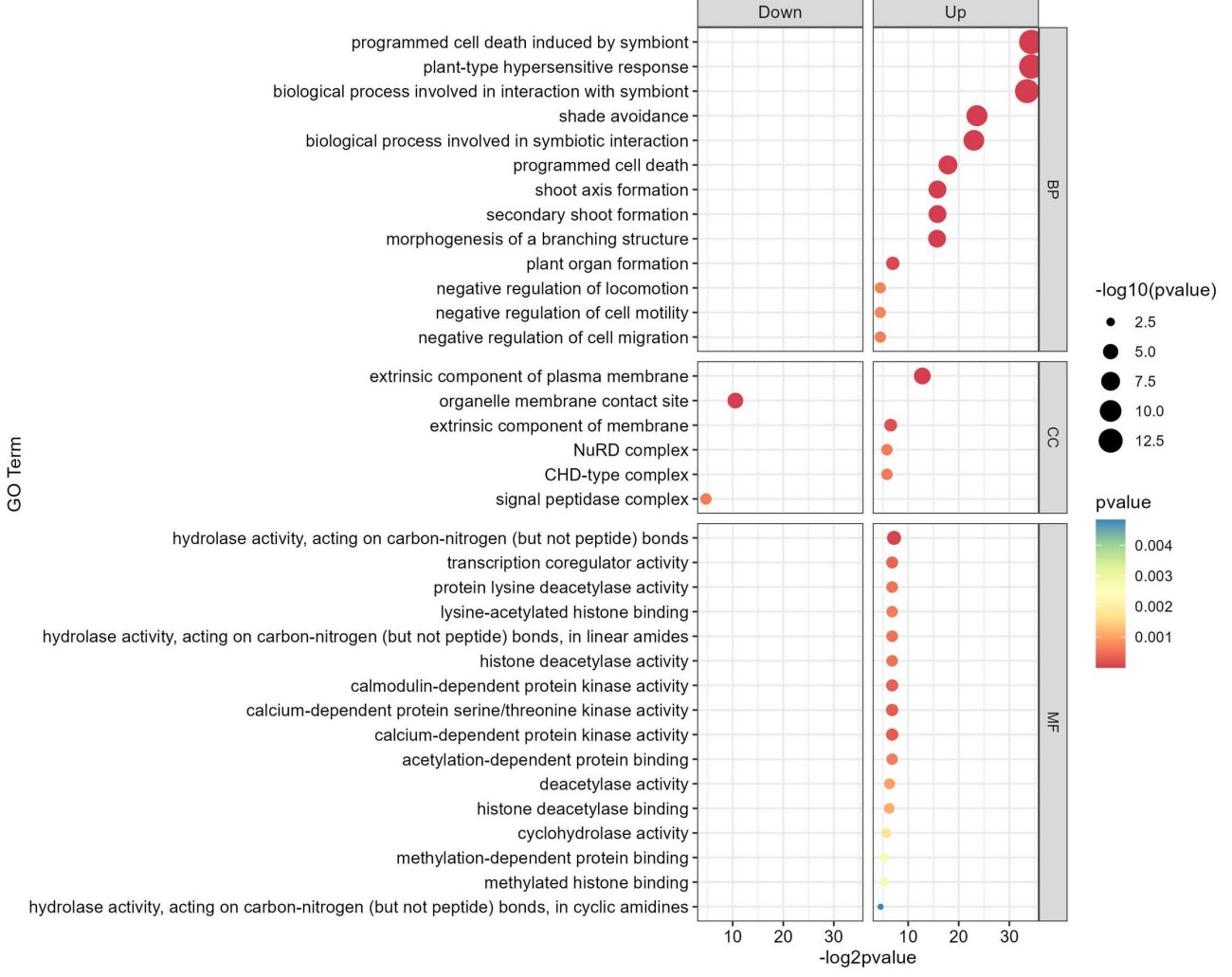

**Fig 5. DEGs functional enrichment analysis.** DEGs enrichment in three GO database samples. Filtering thresholds: p.adjust < 0.05 and p-value< 0.01. Dot size indicates enrichment significance. Use a filtering criterion of p.adjust < 0.2 and p-value < 0.05 to enrich DEGs from three KEGG database samples. Dot size reflects enrichment significance.

(GO: 0051702), shade avoidance (GO: 0009641), symbiotic interaction (GO: 0044403), programmed cell death (GO: 0012501), secondary shoot formation (GO: 0010223), shoot axis formation (GO: 0010346), morphogenesis of branched structures (GO: 0001763), plant organ development (GO: 1905393), negative regulation of cell migration (GO: 0030336), negative regulation of locomotion (GO: 0040013), and negative regulation of cell motility (GO: 2000146). In cellular components, GO terms included the extrinsic component of the plasma membrane (GO: 0019897), membrane extrinsic components (GO: 0019898), NuRD complex (GO: 0016581), CHD- type complex (GO: 0090545), organelle membrane contact sites (GO: 0044232), and signal peptidase complex (GO: 0005787). For molecular functions, enriched GO terms involved hydrolase activity on carbon- nitrogen bonds (excluding peptides) (GO: 0016810), calcium- dependent serine/ threonine kinase activity (GO: 0009931), calcium- dependent kinase activity (GO: 0010857), calmodulin- dependent kinase activity (GO: 0004683), transcription coregulator activity (GO: 0003712), hydrolase activity on carbon- nitrogen bonds in linear amides (GO: 0016811), histone deacetylase activity (GO: 0004407), protein lysine deacetylase activity

(GO: 0033558), lysine- acetylated histone binding (GO: 0070577), acetylation- dependent protein binding (GO: 0140033), deacetylase activity (GO: 0019213), histone deacetylase binding (GO: 0042826), cyclohydrolase activity (GO: 0019238), methylated histone binding (GO: 0035064), methylation- dependent protein binding(GO: 0140034), and hydrolase activity on cyclic amidines (GO: 0016814), as shown in **Fig 5**.

### 3.5. Association of DEGs with growth characters across two soybean cultivars

Mantel tests were employed to evaluate the correlations between significant DEGs and growth characteristics, encompassing plant height, photosynthesis-related attributes: net photosynthetic rate, transpiration rate, intercellular CO2 concentration, stomatal conductance, and leaf-related traits: total chlorophyll content, chlorophyll A, chlorophyll B, and the chlorophyll A/B ratio (**Fig 6b**). The findings indicated that the upregulated and downregulated DEGs were strongly associated with two cultivar characteristics. The heatmap visualization illustrates Pearson correlation coefficients between each important DEG and plant characteristic indicators, revealing unique groups of associated genes (**Fig 6c**). Thirty-seven upregulated DEG exhibited robust positive associations with photosynthesis-related characteristics. A subset of DEGs had a strong negative correlation with ER, indicating potential functions for these genes in preserving membrane stability in shaded environments. Among all DEGs, 28 up-regulated DEGs exhibited a significant negative correlation with

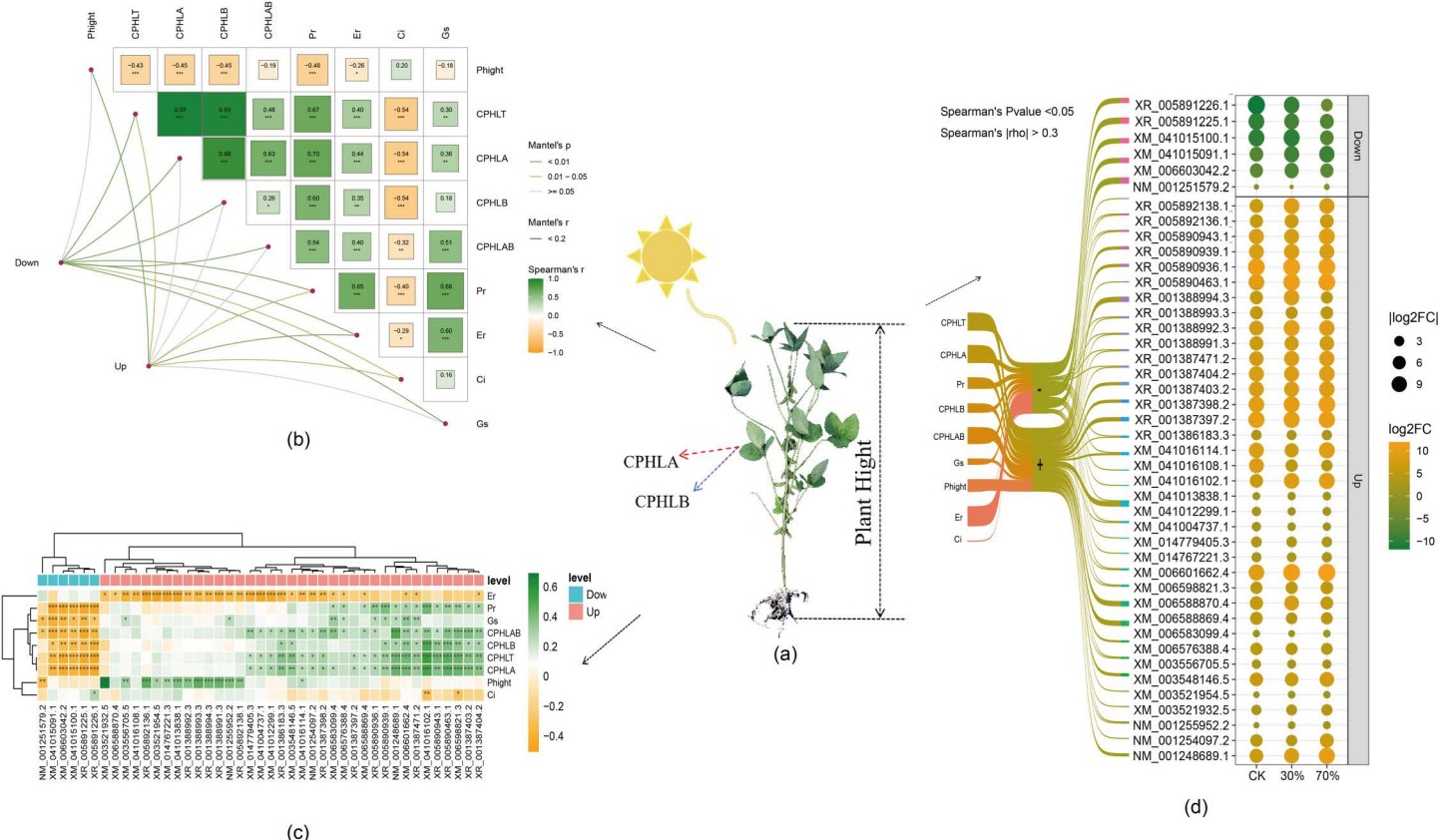

**Fig 6. Association analysis between key DEGs and soybean phenotypes.** (a) Correlations between key DEGs and soybean phenotype. (**b,** and c) Correlation analysis between soybean phenotypes and DEGs functionally annotated to five biological processes. The green line represents up-regulated, and line thickness indicates the correlation strength. (d) A Sankey diagram was generated to visualize the connections between soybean phenotypic traits, differentially expressed genes (DEGs).

estrogen receptor (ER), 2 DEGs (XM_041016102.1 and XM_006598821.3) displayed a negative correlation with crown index (CI), and 12 DEGs (XM_041016114.1, XM_006588870.4, XM_003556705.5, XM_041016108.1, XR_005892136.1, XM_003521954.5, XM_014767221.3, XM_041013838.1, XR_001388992.3, XR_001388993.3, XR_001388994.3, XR_001388991.3, NM_001255952.2, XR_005892138.1) were positively correlated with plant height. The six down-regulated DEGs exhibited a significant negative correlation with photosynthesis-related traits (NM_001251579.2, XM_041015091.1, XM_006603042.2, XM_041015100.1, XR_005891225.1, XR_005891226.1). Additionally, one down-regulated gene (XR_005891226.1) demonstrated a significant negative correlation with plant height (**Fig 6c**). The Sankey diagram in **Fig 6d** depicts the interconnections between physiological parameters, their associations with DEGs, DEG functions, and DEG regulation (upregulation and downregulation) under shade conditions across three treatment levels. This comprehensive visualization illustrates the relationships between physiological markers and DEGs during shade stress across the treatment gradient. Moreover, it reveals the strong correlation between physiological changes and gene expression alterations across multiple biological pathways throughout the shade exposure period.

## 4. Discussion

Multiple investigations have validated that certain soybean cultivars exhibit distinct adaptive responses to shaded environments [27,28]. Shade-tolerant cultivars possess a minimal or absent stratum corneum, extensive leaf area, abundant foliage, and a limited number of stomata and chloroplasts [29,30]. Previous studies demonstrate that under the conventional intercropping system of corn and soybean, soybean experienced a diminished light availability, decreasing by 30% to 50% relative to standard soybean cultivation [5].Our investigation elucidated the physiological and transcriptional responses of soybean cultivars to varying degrees of shade stress. Physiological assessments coupled with RNA-Seq analysis of Guru and Heinong 53 cultivars revealed that shade tolerance mechanisms exhibit genotypic specificity and are dependent on shade intensity gradients. There were many DEGs reported by the transcriptome study when comparing the treatments; for example, there were more upregulated and downregulated genes under 70% shade than 30% shade. Light signaling, photosynthesis, and shadow avoidance responses were the most common GO categories among the differentially expressed genes (DEGs). The "shade avoidance" (GO:0009641) and "plant organ formation" (GO:1905393) biological processes were dramatically affected. Guru, a shade-tolerant cultivar, can keep photosynthetic efficiency in light-limited conditions by overexpressing key DEGs associated with photosynthesis and carbon metabolism when exposed to greater shade.

Consistent with the transcriptome results, the phenotypic and physiological data highlight the distinct strategies employed by the two varieties. The Guru showed a more consistent plant height in darkened conditions, suggesting that resources were efficiently distributed between vertical growth and the extension of leaf area. Upon analysis, Heinong 53 demonstrated significant stem elongation under reduced light conditions, a characteristic shade avoidance response that potentially compromises total biomass production. This finding aligns with previous studies indicating that exaggerated elongation can negatively impact carbon partitioning to reproductive organs and impair chlorophyll biosynthesis pathways.

### Chlorophyll content and photosynthetic adaptations

The two cultivars' differences in chlorophyll concentration and photosynthetic characteristics reveal unique adaptation mechanisms to shade stress. Guru exhibited consistently high chlorophyll content and photosynthetic efficiency under moderate (30%) and severe (70%) shading circumstances, indicating enhanced pathways for light absorption and utilization in the shade-tolerant genotype. The stable chlorophyll A/B ratio in Guru indicates effective adaptation to shady habitats, as this ratio is a vital indicator of photosystem alterations in reaction to varying light intensities. Conversely, Heinong 53 exhibited a decline in both chlorophyll concentration and photosynthetic rate under increased shadow conditions. The noted reductions correspond with the downregulation of genes linked to photosynthesis in Heinong 53, as demonstrated by transcriptome analysis. The diminished stomatal conductance (Gs) and transpiration rate (Er) in Heinong 53 under

70% shadow may impede CO2 absorption, thereby exacerbating the decline in photosynthesis. The chlorophyll content was directly correlated with the shade stress experienced by plants [31]. Carotenoids in green leaves facilitate effective photosynthesis, eliminate reactive oxygen species, and safeguard chlorophyll from photooxidation [32,33].

Shade-tolerant plants possess dense foliage, extensive leaf surface areas, an absent or minimal stratum corneum, fewer stomata, and chloroplasts [34,35]. Plants exhibiting greater shade tolerance possess elevated levels of chlorophyll b and a diminished chlorophyll a/b ratio [35]. In low light conditions, plants exhibiting a reduced chlorophyll a/b ratio and elevated chlorophyll content demonstrated increased photosynthetic activity [36,37]. The elevated chlorophyll content and transcript levels of chlorophyll a/b-binding protein genes indicated that soybean enhanced the transcription of light reaction-related genes to adapt to low-light circumstances for optimizing the capture of limited light resources [28,38,39]. The reduction in cell quantity and low light conditions diminished leaf thickness by limiting anticlinal cell expansion rates, an impact that was preceded by a decrease in cell division, resulting in one fewer layer of palisade cells [40].

## Molecular mechanisms of shade tolerance

The Mantel test revealed significant associations between the differentially expressed genes and key physiological variables, such as plant height, photosynthetic rate, and chlorophyll concentration. Several elevated DEGs in Guru under high shade circumstances demonstrated a favorable connection with photosynthesis-related characteristics, suggesting their potential role in enhancing shade tolerance. The genes linked to "calcium-dependent protein kinase activity" (GO:0009931) and "transcription coregulator activity" (GO:0003712) exhibited elevated expression in Guru, potentially contributing to improved photoprotection and the regulation of photosynthetic processes. The upregulation of genes associated with calcium-dependent protein kinase activity in Guru represents a critical adaptive mechanism in shade tolerance. Calcium-dependent protein kinases (CDPKs) function as calcium sensors and play pivotal roles in stress signal transduction [24]. Under shade conditions, the altered red:far-red light ratio is perceived by phytochromes, which triggers calcium influx into the cytosol. In Guru, enhanced CDPK activity likely facilitates activation of the CBL-CIPK signaling network, which regulates ion channels and transporters to maintain cellular homeostasis under shade stress [23,24]. Additionally, CDPKs may contribute to the phosphorylation of photosystem II proteins, particularly the D1 protein, enhancing repair mechanisms during photoinhibition and optimizing proton gradients for energy conservation under limited light conditions [22].

The elevated expression of genes related to transcription coregulator activity in Guru suggests sophisticated transcriptional reprogramming during shade adaptation. These coregulators likely modify the activity of transcription factors involved in photomorphogenesis, particularly PIF repression pathways, which prevent excessive stem elongation under shade [22]. Enhanced activity of HY5 transcription factors, which promotes photosynthetic gene expression and chloroplast development, may also contribute to Guru's shade tolerance. Furthermore, modulation of JA-mediated signaling pathways through interaction with JAZ proteins balances growth with defense responses, optimizing resource allocation by prioritizing photosynthetic efficiency over shade avoidance responses [21].

## Stress response and energy conservation strategies

The diminishment of gene expression associated with shade avoidance in Guru, including those about programmed cell death and hypersensitive reactions, may facilitate a decrease in energy consumption after prolonged shade exposure. Under prolonged shade, Guru appears to deactivate the MAPK cascade components that typically initiate programmed cell death during stress and ROS-generating enzymes like NADPH oxidases, reducing cellular oxidative damage. Additionally, suppressing metacaspases and other proteases involved in executing cell death programs likely contributes to Guru's energy conservation strategy under shade conditions. This contrasts with Heinong 53, where the activation of these genes in response to shade stress may indicate maladaptive stress responses. In the shade-sensitive cultivar, the upregulation of respiratory burst oxidase homologs that generate stress-induced ROS, ER stress-related genes triggering the unfolded protein response, and autophagy-related genes may lead to excessive cellular degradation. In Heinong

53, this suboptimal response redirects metabolic resources from photosynthetic apparatus maintenance toward stress-responsive mechanisms, thus diminishing its capacity to sustain productivity under light-limited conditions.

The observed positive correlation between specific DEGs and photosynthetic metrics in Guru indicates photosynthetic machinery optimization under shade stress through several mechanisms: upregulated expression of genes encoding light-harvesting complex proteins particularly those affiliated with photosystem I thereby enhancing light-harvesting efficiency; upregulation of cytochrome b6f complex components optimizing electron transport chain function under limited light; modified stomatal regulation through altered expression of ion channels and transporters balancing gas exchange with water conservation [41]; and increased expression of genes involved in photorespiratory bypasses minimizing energy loss from photorespiration under low light conditions, collectively enhancing quantum efficiency of photosystem II and carbon assimilation rates in Guru.

The molecular insights revealed by this study provide a foundation for targeted breeding approaches to enhance shade tolerance in soybean and potentially other crops. Future research should focus on key regulatory nodes in calcium signaling, transcriptional regulation, and photosynthetic optimization pathways to develop varieties with improved performance under intercropping systems and other shade-prone agricultural practices. Understanding these shade adaptation mechanisms at the molecular level opens new avenues for genetic improvement strategies aimed at sustainable crop production under diverse light environments.

## 5. Conclusion

Our investigation elucidates the mechanistic underpinnings of shade tolerance by characterizing differential responses of two soybean cultivars, Guru and Heinong 53, to graduated levels of shade stress. The shade-tolerant genotype Guru exhibited remarkable physiological plasticity under both moderate (30%) and severe (70%) shade conditions, manifesting enhanced photosynthetic efficiency, elevated chlorophyll content, and an optimized leaf-to-stem resource allocation ratio. These adaptive traits were substantiated by transcriptional modulations that prioritized photosynthetic capacity, light capture efficiency, and shade acclimation over shade avoidance responses. Conversely, the shade-susceptible genotype Heinong 53 demonstrated diminished chlorophyll biosynthesis and photosynthetic rates concurrent with excessive stem elongation when subjected to shade stress, presumably compromising its overall performance. The observed physiological and molecular response divergence illuminates the genetic and phenotypic architecture underlying shade tolerance in soybean. These findings constitute valuable resources for elucidating molecular mechanisms of soybean shade tolerance and offer strategic implications for breeding programs aimed at optimizing soybean productivity in intercropping systems or light-limited environments.

## Supporting information

**S1 Table. RNA-sequencing libraries quality metrics including read counts, GC content, and mapping statistics for all samples.**
(XLSX)

**S2 Table. Detailed quality control measures for RNA libraries, including filtration parameters and alignment percentages.**
(XLSX)

## Acknowledgments

Special thanks to the Soybean intellect design breeding laboratory of Heilongjiang Academy of Agricultural Sciences for providing platform support. Soybean Germplasm Resources Team of the Institute of Crop Sciences, Chinese Academy of Agricultural Sciences for providing soybean germplasm resources.

## Author contributions

**Conceptualization:** Fengyi Zhang, Rongqiang Yuan, Sobhi F. Lamlom, Dai Shi.

**Data curation:** Fengyi Zhang, Rongqiang Yuan, Sobhi F. Lamlom, Dai Shi.

**Formal analysis:** Fengyi Zhang, Rongqiang Yuan, Sobhi F. Lamlom, Dai Shi, Lijuan Qiu.

**Funding acquisition:** Fengyi Zhang, Rongqiang Yuan.

**Investigation:** Runnan Zhou, Dai Shi, Ahmed M. Abdelghany, Lijuan Qiu.

**Methodology:** Runnan Zhou, Xiao Zhu, Dai Shi, Ahmed M. Abdelghany.

**Project administration:** Fengyi Zhang, Runnan Zhou, Xiao Zhu, Xinyue Zhang, Honglei Ren.

**Resources:** Runnan Zhou, Xiao Zhu, Xinyue Zhang, Honglei Ren.

**Software:** Xiao Zhu, Xinyue Zhang, Sobhi F. Lamlom, Jidao Du, Honglei Ren, Lijuan Qiu.

**Supervision:** Fengyi Zhang, Xinyue Zhang, Jidao Du, Honglei Ren, Lijuan Qiu.

**Validation:** Sobhi F. Lamlom, Ahmed M. Abdelghany, Jidao Du, Lijuan Qiu.

**Visualization:** Sobhi F. Lamlom, Ahmed M. Abdelghany, Jidao Du, Honglei Ren, Lijuan Qiu.

**Writing – original draft:** Fengyi Zhang, Sobhi F. Lamlom, Ahmed M. Abdelghany, Honglei Ren.

**Writing – review & editing:** Sobhi F. Lamlom, Ahmed M. Abdelghany, Jidao Du, Honglei Ren.

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
