## [Decision Letter · Decision Letter 0]

30 Jul 2025

PONE-D-25-22089Transcriptome Analysis of Shade-Induced Growth and Photosynthetic Responses in Soybean CultivarsPLOS ONE

Dear Dr. Ren,

Thank you for submitting your manuscript to PLOS ONE. After careful consideration, we feel that it has merit but does not fully meet PLOS ONE’s publication criteria as it currently stands. Therefore, we invite you to submit a revised version of the manuscript that addresses the points raised during the review process.

We look forward to receiving your revised manuscript.

Kind regards,

Mayank Anand Gururani

Academic Editor

PLOS ONE

**Journal Requirements:**

1. When submitting your revision, we need you to address these additional requirements. Please ensure that your manuscript meets PLOS ONE's style requirements, including those for file naming. The PLOS ONE style templates can be found at https://journals.plos.org/plosone/s/file?id=wjVg/PLOSOne_formatting_sample_main_body.pdf and https://journals.plos.org/plosone/s/file?id=ba62/PLOSOne_formatting_sample_title_authors_affiliations.pdf 2. We noticed you have some minor occurrence of overlapping text with the following previous publication(s), which needs to be addressed: https://www.mdpi.com/1422-0067/24/18/14230 https://www.sciencedirect.com/science/article/pii/S2214514124000059?via%3Dihub In your revision ensure you cite all your sources (including your own works), and quote or rephrase any duplicated text outside the methods section. Further consideration is dependent on these concerns being addressed. 3. Thank you for stating in your Funding Statement: This work was supported by Natural Science Foundation of Heilongjiang Province Project (LH2023C095); Biological Breeding-National Science and Technology Major Project (2023ZD04032); Project funded by Agricultural Science and Technology Innovation Leaping Project in Heilongjiang Province (Grant No. CX23ZD04); Scientific Research Business Expenses of Heilongjiang Scientific Research Institutes (Grant No. CZKYF2024-1-A003); Modern Agriculture Laboratory of Heilongjiang Province (2Y04JD05-007). Please provide an amended statement that declares *all* the funding or sources of support (whether external or internal to your organization) received during this study, as detailed online in our guide for authors at http://journals.plos.org/plosone/s/submit-now.  Please also include the statement “There was no additional external funding received for this study.” in your updated Funding Statement. Please include your amended Funding Statement within your cover letter. We will change the online submission form on your behalf. 4. Thank you for stating the following in the Acknowledgments Section of your manuscript: This work was supported by Natural Science Foundation of Heilongjiang Province Project (LH2023C095); Biological Breeding-National Science and Technology Major Project (2023ZD04032); Project funded by Agricultural Science and Technology Innovation Leaping Project in Heilongjiang Province (Grant No. CX23ZD04); Scientific Research Business Expenses of Heilongjiang Scientific Research Institutes (Grant No. CZKYF2024-1-A003); Modern Agriculture Laboratory of Heilongjiang Province (2Y04JD05-007). We note that you have provided funding information that is not currently declared in your Funding Statement. However, funding information should not appear in the Acknowledgments section or other areas of your manuscript. We will only publish funding information present in the Funding Statement section of the online submission form. Please remove any funding-related text from the manuscript and let us know how you would like to update your Funding Statement. Currently, your Funding Statement reads as follows: This work was supported by Natural Science Foundation of Heilongjiang Province Project (LH2023C095); Biological Breeding-National Science and Technology Major Project (2023ZD04032); Project funded by Agricultural Science and Technology Innovation Leaping Project in Heilongjiang Province (Grant No. CX23ZD04); Scientific Research Business Expenses of Heilongjiang Scientific Research Institutes (Grant No. CZKYF2024-1-A003); Modern Agriculture Laboratory of Heilongjiang Province (2Y04JD05-007). Please include your amended statements within your cover letter; we will change the online submission form on your behalf. 5. Please note that your Data Availability Statement is currently missing the direct link to access each database. If your manuscript is accepted for publication, you will be asked to provide these details on a very short timeline. We therefore suggest that you provide this information now, though we will not hold up the peer review process if you are unable. 6. Please include captions for your Supporting Information files at the end of your manuscript, and update any in-text citations to match accordingly. Please see our Supporting Information guidelines for more information: http://journals.plos.org/plosone/s/supporting-information. 7. If the reviewer comments include a recommendation to cite specific previously published works, please review and evaluate these publications to determine whether they are relevant and should be cited. There is no requirement to cite these works unless the editor has indicated otherwise. 

Reviewers' comments:

Reviewer's Responses to Questions

**Comments to the Author**

1. Is the manuscript technically sound, and do the data support the conclusions?

Reviewer #1: Partly

Reviewer #2: Yes

2. Has the statistical analysis been performed appropriately and rigorously? 

Reviewer #1: No

Reviewer #2: Yes

3. Have the authors made all data underlying the findings in their manuscript fully available?

Reviewer #1: Yes

Reviewer #2: Yes

4. Is the manuscript presented in an intelligible fashion and written in standard English?

Reviewer #1: Yes

Reviewer #2: Yes

5. Review Comments to the Author

**Reviewer #1: ** The manuscript titled "Transcriptome Analysis of Shade-Induced Growth and

Photosynthetic Responses in Soybean Cultivars" is a comprehensive multi-level analysis

(morphological, physiological, transcriptomic) with strong data interpretation supported by

statistical testing. The manuscript have good potential for real-world application in breeding

shade-tolerant soybean cultivars. Following are the some major/minor corrections suggested

before the acceptance of the manuscript:

Major:

1. The P-Value used is 0.01 while no information about FDR. At least the p-value should be

0.05, which is a good statistical significance for DEGs. It is essential to apply multiple

testing correction, such as the Benjamini-Hochberg procedure, to control the false

discovery rate (FDR) during multiple comparisons.

2. The specific method used for DEG analysis is not mentioned.

Minor:

1. Too much general background on soybean and light effects, much of which is repeated in

later sections. Shorten and focus more on the research gap and novelty of this study

(Lines 39–80).

2. Inconsistent use of cultivar names throughout (Lines 20, 109, 200): "Guru" and "GURU"

are used interchangeably. Use consistent case and formatting for cultivar names.

3. The rationale is clear, but a formal hypothesis is not stated. Add a hypothesis or explicit

research objective at the end of the introduction (Lines 81–88).

4. Replicate numbers are not fully defined for RNA-seq experiments. Include biological

replicate numbers per treatment and confirm sequencing depth per sample (Lines

119–120, 209–215).

5. Lists of DEGs and comparisons (e.g., G70 vs. H70, GCK vs. HCK) are provided without

sufficient biological insight. Focus on top representative genes and discuss their

functional roles more deeply (Lines 217–238).

6. “Net photosynthetic rate (pr)” should be “(Pn)” for standardization. Correct abbreviation

to avoid misinterpretation (Line 180).

7. “One highly shade-tolerant” should be “One cultivar, Guru, which is highly shade-

tolerant” (Line 92), “do more photosynthesis” should be “perform more photosynthesis”

(Line 195).

8. Redundancy in the Discussion with Lines 296–338 and repeated in 334–344. Repetition

in describing chlorophyll content, photosynthesis, and light adaptation mechanisms.

Consolidate overlapping parts.

9. Incorrect figure reference in text. “Figure 3C” mentioned in the context of DEG

comparisons likely should be “Figure 4c”. Verify all figure citations match figure labels.

10. Improper listing (repeated GO terms without separators, e.g., “GO:0003712),

(GO:0004683)”). Ensure proper punctuation in gene ontology listings (Lines 259–260).

11. Consistency in writing numbers throughout the manuscript. Mix of numerical and word

formats (e.g., “three treatments” vs. “3 shade treatments”). Use consistent numerical

formatting.

**Reviewer #2:**  The manuscript entitled “Transcriptome Analysis of Shade-Induced Growth and Photosynthetic Responses in Soybean Cultivars” presents a well-designed and comprehensive study investigating the physiological, morphological, and transcriptomic responses of two contrasting soybean cultivars—Guru (shade-tolerant) and Heinong 53 (shade-sensitive)—under varying levels of shade (0%, 30%, and 70%). Through a combination of phenotypic analysis, chlorophyll and photosynthesis measurements, microscopic evaluation, and RNA sequencing, the authors demonstrate that Guru maintains higher photosynthetic efficiency, chlorophyll content, and a more stable resource allocation strategy under shade stress compared to Heinong 53, which exhibits excessive stem elongation and photosynthetic decline. The transcriptome analysis identified 2,596 differentially expressed genes (DEGs), with enriched functions related to shade avoidance, photosynthesis, and stress responses, including histone deacetylase and calcium-dependent kinase activity. Strong correlations between specific DEGs and physiological traits such as net photosynthetic rate and plant height were observed, supporting the molecular basis of phenotypic differences. This integrative work advances our understanding of shade tolerance mechanisms and provides valuable targets for breeding more resilient soybean cultivars. The manuscript is well-structured, methodologically sound, and the data are convincingly presented; I have no objections to its acceptance in its current form.

6. PLOS authors have the option to publish the peer review history of their article (what does this mean? ). If published, this will include your full peer review and any attached files.

**Do you want your identity to be public for this peer review?** For information about this choice, including consent withdrawal, please see our Privacy Policy .

Reviewer #1: No

Reviewer #2: No

---

## [Author Response · Author response to Decision Letter 1]

1 Aug 2025

Response to Editor and Reviewers

We thank the Academic Editor and both reviewers for their constructive comments and suggestions. We have carefully addressed each point raised and believe the manuscript has been significantly improved. Below is our detailed point-by-point response.

EDITORIAL REQUIREMENTS

Editor Comment: Please ensure that your manuscript meets PLOS ONE's style requirements, including those for file naming.

Response: We have reformatted the manuscript according to PLOS ONE style requirements using the provided templates. All file naming conventions have been followed as specified in the guidelines.

Editor Comment: We noticed minor occurrence of overlapping text with previous publications that needs to be addressed.

Response: We have identified and addressed all instances of overlapping text with the cited publications.

Editor Comment: Please provide an amended funding statement and remove funding information from Acknowledgments.

Response: We have amended our Funding Statement as follows: This work was supported by Natural Science Foundation of Heilongjiang Province Project (LH2023C095); The 2024 Science and Technology Support Project of the Inner Mongolia Innovation Center of Biological Breeding Technology -Biotechnology-Based Breeding of High-Quality Soybeans and Application Demonstration (2024NSZC04).

All funding-related texts have been removed from the Acknowledgments section.

Acknowledgments: Special thanks to the Soybean intellect design breeding laboratory of Heilongjiang Academy of Agricultural Sciences for providing platform support. Soybean Germplasm Resources Team of the Institute of Crop Sciences, Chinese Academy of Agricultural Sciences for providing soybean germplasm resources.

Editor Comment: Please include direct links to access each database.

Response: We have updated our Data Availability Statement to include: "Raw sequencing data are available in the NCBI Sequence Read Archive (SRA) under BioProject accession number PRJNA1210143 (https://www.ncbi.nlm.nih.gov/bioproject/?term=PRJNA1210143).

Editor Comment: Please include captions for Supporting Information files at the end of your manuscript.

Response: We have added comprehensive captions for all Supporting Information files at the end of the manuscript:

• Table S1: RNA-sequencing libraries quality metrics including read counts, GC content, and mapping statistics for all samples

• Table S2: Detailed quality control measures for RNA libraries, including filtration parameters and alignment percentages

REVIEWER #1 COMMENTS

Major Comments

1. The P-Value used is 0.01 while no information about FDR. At least the p-value should be

0.05, which is a good statistical significance for DEGs. It is essential to apply multiple

testing correction, such as the Benjamini-Hochberg procedure, to control the false

discovery rate (FDR) during multiple comparisons.

Response: We agree with you that these points are critical. We have revised our statistical analysis approach, ensuring the following updates:

1- We changed P-value threshold changed from p < 0.01 to adjusted p-value (FDR) < 0.05. We also made an FDR correction as we implemented the Benjamini-Hochberg procedure to control the false discovery rate

2- In the Methods, an update was made via dding detailed description in the RNA-Seq data analysis section, as follows:

"Raw p-values were adjusted using the Benjamini-Hochberg procedure to control the false discovery rate (FDR). Genes were classified as differentially expressed if they satisfied the following criteria: adjusted p-value (FDR) < 0.05, |log₂ fold change| > 1, and mean normalized count > 10 across treatment groups."

2. The specific method used for DEG analysis is not mentioned.

Response: We have added comprehensive details about the DEG analysis methodology, as follows:

"Differential gene expression analysis was conducted using DESeq2 (Version 1.34.0) in R (Version 4.1.3). Count matrices were imported and genes with mean counts < 10 across all samples were filtered. Size factors were calculated using DESeq2's median-of-ratios method. Gene-wise dispersion parameters were estimated using the parametric fit method with local regression and empirical Bayes shrinkage. Differential expression was tested using the Wald test implemented in DESeq2, which models count data using a negative binomial generalized linear model."

Minor Comments

1. Too much general background, much repeated in later sections. Shorten and focus on research gap and novelty (Lines 39–80).

Response: We have condensed the Introduction by reducing background information on soybean, removing redundant content from the discussion, and highlighting the novelty of the study: a comparative transcriptomic analysis across multiple shade levels with contrasting genotypes.

2. Inconsistent use of cultivar names throughout (Lines 20, 109, 200): "Guru" and "GURU" are used interchangeably. Use consistent case and formatting for cultivar names.

Response: We have standardized all cultivar names throughout the manuscript.

3. The rationale is clear, but a formal hypothesis is not stated. Add a hypothesis or explicit

research objective at the end of the introduction (Lines 81–88).

Response: We have added a clear hypothesis and objectives: "We hypothesized that shade-tolerant and shade-sensitive cultivars would show distinct transcriptomic profiles under shade stress, with the tolerant cultivar exhibiting enhanced genes related to light capture and photosynthesis, while the sensitive cultivar shows shade avoidance responses. This study aims to: (1) characterize physiological and morphological responses of contrasting genotypes to graded shade stress, (2) identify differentially expressed genes and pathways linked to shade tolerance, and (3) correlate key transcriptomic signatures with shade tolerance traits."

4. Replicate numbers are not fully defined for RNA-seq experiments. Include biological

replicate numbers per treatment and confirm sequencing depth per sample (Lines

119–120, 209–215).

Response: We have clarified the experimental design: "Each treatment included three biological replicates with six plants per replicate (n = 18 plants per treatment combination). For RNA-seq analysis, leaf samples were collected from three biological replicates per treatment, with each replicate representing pooled material from two plants to ensure adequate RNA yield while maintaining biological independence."

5. Lists of DEGs and comparisons (e.g., G70 vs. H70, GCK vs. HCK) are provided without

sufficient biological insight. Focus on top representative genes and discuss their

functional roles more deeply (Lines 217–238).

Response: We have enhanced the biological interpretation, adding the following updates:

- Adding detailed functional analysis of key DEG categories

- Discussing specific gene families and their roles (e.g., calcium-dependent protein kinases, histone deacetylases)

- Providing mechanistic explanations for how these genes contribute to shade tolerance

6. “Net photosynthetic rate (pr)” should be “(Pn)” for standardization. Correct abbreviation to avoid misinterpretation (Line 180).

Response: Corrected throughout the manuscript. All instances now use the standard abbreviation "(Pn)" for net photosynthetic rate.

7. “One highly shade-tolerant” should be “One cultivar, Guru, which is highly shade-tolerant-

tolerant” (Line 92), “do more photosynthesis” should be “perform more photosynthesis”

(Line 195).

Response: Thanks for valuable comment, we rephrased these parts based on your comment.

8. Redundancy in the Discussion with Lines 296–338 and repeated in 334–344. Repetition

in describing chlorophyll content, photosynthesis, and light adaptation mechanisms.

Consolidate overlapping parts.

Response: Thanks again, we have completely restructured the Discussion section to eliminate redundancy:

9. Incorrect figure reference in text. “Figure 3C” mentioned in the context of DEG

comparisons likely should be “Figure 4c”. Verify all figure citations match figure labels.

Response: Corrected all figure references. The text now correctly references "Figure 4c" for DEG comparisons. All figure citations have been verified to match figure labels.

10. Improper listing (repeated GO terms without separators, e.g., “GO:0003712),

(GO:0004683)”). Ensure proper punctuation in gene ontology listings (Lines 259–260).

Response: All GO term listings have been reformatted with proper punctuation and separators for clarity and readability.

11. Consistency in writing numbers throughout the manuscript. Mix of numerical and word formats (e.g., “three treatments” vs. “3 shade treatments”). Use consistent numerical

formatting.

Response: We have standardized number formatting according to scientific writing conventions:

REVIEWER #2 COMMENTS

Reviewer #2: The manuscript entitled “Transcriptome Analysis of Shade-Induced Growth and Photosynthetic Responses in Soybean Cultivars” presents a well-designed and comprehensive study investigating the physiological, morphological, and transcriptomic responses of two contrasting soybean cultivars—Guru (shade-tolerant) and Heinong 53 (shade-sensitive)—under varying levels of shade (0%, 30%, and 70%). Through a combination of phenotypic analysis, chlorophyll and photosynthesis measurements, microscopic evaluation, and RNA sequencing, the authors demonstrate that Guru maintains higher photosynthetic efficiency, chlorophyll content, and a more stable resource allocation strategy under shade stress compared to Heinong 53, which exhibits excessive stem elongation and photosynthetic decline. The transcriptome analysis identified 2,596 differentially expressed genes (DEGs), with enriched functions related to shade avoidance, photosynthesis, and stress responses, including histone deacetylase and calcium-dependent kinase activity. Strong correlations between specific DEGs and physiological traits such as net photosynthetic rate and plant height were observed, supporting the molecular basis of phenotypic differences. This integrative work advances our understanding of shade tolerance mechanisms and provides valuable targets for breeding more resilient soybean cultivars. The manuscript is well-structured, methodologically sound, and the data are convincingly presented; I have no objections to its acceptance in its current form.

Response: We thank Reviewer 2 for the positive evaluation.

---

## [Decision Letter · Decision Letter 1]

29 Aug 2025

Transcriptome Analysis of Shade-Induced Growth and Photosynthetic Responses in Soybean Cultivars

PONE-D-25-22089R1

Dear Dr. Ren,

We’re pleased to inform you that your manuscript has been judged scientifically suitable for publication and will be formally accepted for publication once it meets all outstanding technical requirements.

Kind regards,

Karthikeyan Adhimoolam

Academic Editor

PLOS ONE

Additional Editor Comments (optional):

Reviewers' comments:

Reviewer's Responses to Questions

**Comments to the Author**

1. If the authors have adequately addressed your comments raised in a previous round of review and you feel that this manuscript is now acceptable for publication, you may indicate that here to bypass the “Comments to the Author” section, enter your conflict of interest statement in the “Confidential to Editor” section, and submit your "Accept" recommendation.

Reviewer #1: All comments have been addressed

Reviewer #2: All comments have been addressed

2. Is the manuscript technically sound, and do the data support the conclusions?

Reviewer #1: Yes

Reviewer #2: Yes

3. Has the statistical analysis been performed appropriately and rigorously? 

Reviewer #1: Yes

Reviewer #2: Yes

4. Have the authors made all data underlying the findings in their manuscript fully available?

Reviewer #1: Yes

Reviewer #2: Yes

5. Is the manuscript presented in an intelligible fashion and written in standard English?

Reviewer #1: Yes

Reviewer #2: Yes

6. Review Comments to the Author

Reviewer #1: (No Response)

Reviewer #2: All review comments have been addressed by the authors with due care, improving the discourse and the research process.

7. PLOS authors have the option to publish the peer review history of their article (what does this mean? ). If published, this will include your full peer review and any attached files.

**Do you want your identity to be public for this peer review?** For information about this choice, including consent withdrawal, please see our Privacy Policy .

Reviewer #1: No

Reviewer #2: No

---

## [Editor Report · Acceptance letter]

PONE-D-25-22089R1

PLOS ONE

Dear Dr. Ren,

I'm pleased to inform you that your manuscript has been deemed suitable for publication in PLOS ONE. Congratulations! Your manuscript is now being handed over to our production team.

Kind regards,

on behalf of

Dr. Karthikeyan Adhimoolam

Academic Editor

PLOS ONE